# THINK-THEN-REACT: TOWARDS UNCONSTRAINED HUMAN ACTION-TO-REACTION GENERATION

**Wenhui Tan, Boyuan Li, Chuhao Jin, Wenbing Huang, Xiting Wang & Ruihua Song**[*]
Gaoling School of Artificial Intelligence
Renmin University of China
Beijing, China
{tanwenhui404,liboyuan,jinchuhao,hwenbing,xitingwang,rsong}@ruc.edu.cn

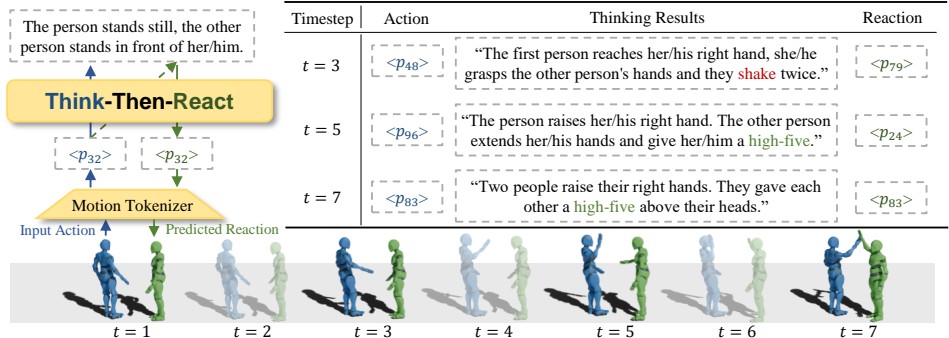

Figure 1: Given a human action as input, our Think-Then-React model first **thinks** by generating an action description and reasons out a reaction prompt. It then **reacts** to the action based on the results of this thinking process. TTR reacts in a real-time manner at every timestep and periodically re-thinks at specific interval (every two timesteps in the illustration) to mitigate accumulated errors.

## ABSTRACT

Modeling human-like action-to-reaction generation has significant real-world applications, like human-robot interaction and games. Despite recent advancements in single-person motion generation, it is still challenging to well handle action-to-reaction generation, due to the difficulty of directly predicting reaction from action sequence without prompts, and the absence of a unified representation that effectively encodes multi-person motion. To address these challenges, we introduce Think-Then-React (TTR), a large language-model-based framework designed to generate human-like reactions. First, with our fine-grained multimodal training strategy, TTR is capable to unify two processes during inference: a **thinking** process that explicitly infers action intentions and reasons corresponding reaction description, which serve as semantic prompts, and a **reacting** process that predicts reactions based on input action and the inferred semantic prompts. Second, to effectively represent multi-person motion in language models, we propose a unified motion tokenizer by decoupling egocentric pose and absolute space features, which effectively represents action and reaction motion with same encoding. Extensive experiments demonstrate that TTR outperforms existing baselines, achieving significant improvements in evaluation metrics, such as reducing FID from 3.988 to 1.942.

## 1 INTRODUCTION

Predicting human reaction to human action in real world scenario is an online and unconstrained task, i.e., future states and text prompts are inaccessible, and it has board applications in virtual reality, human-robot interaction and gaming. Recently, significant advancements have been achieved

---

[*]Corresponding author: Ruihua Song (rsong@ruc.edu.cn)

in the domain of human motion generation especially single-person motion generation, conditioned on text prompts (Guo et al., 2024; 2022b; Zhang et al., 2023) and action labels (Xu et al., 2023; Guo et al., 2020). Leveraging well-annotated human motion datasets (Xu et al., 2024a; Guo et al., 2022a; Liu et al., 2020; Plappert et al., 2016), these models employ various generative frameworks, such as Diffusion Models (Ho et al., 2020; Liang et al., 2024; Zhang et al., 2022), Variational Autoencoders (VAEs) (Kingma, 2013; Petrovich et al., 2021b), and Generative Adversarial Networks (GANs) (Goodfellow et al., 2014; Men et al., 2022), to capture cross-modality distributions for better motion generation. Furthermore, Large Language Models (LLMs) have been applied to human motion generation, demonstrating superior performance (Jiang et al., 2023; Zhang et al., 2024).

However, generating human reaction in multi-person scenario presents a more challenging task due to two primary factors. First, directly predicting reaction from action sequence is a difficult task with instability. As shown in Figure 1, given first two action steps, it is ambiguous to distinguish whether the action is "shake hand" or "high five", and this would lead to accumulated error to consequent predicted reactions. Second, dissimilar to single-person motion representation that can adopt an egocentric view, representing human motion in multi-person scenario necessitates both egocentric and absolute information.

Several works have focused on human interaction domain. For instance, InterFormer (Chopin et al., 2023) proposes injecting human skeleton priors into transformer attention layers for effective spatial modeling. InterGen (Liang et al., 2024) introduces a mutual attention mechanism within diffusion process for joint action-reaction generation. However, these methods are not directly applicable to real-world applications, as they rely on extra prompts to condition the generation process. ReGen-Net (Xu et al., 2024b), similar to our approach, acknowledges the online and unprompted nature of reaction generation, and proposes a diffusion-based model for online reaction generation. It observes that given the action's intention as a condition explicitly, the model can achieve superior performance compared to unprompted settings, highlighting the necessity of understanding interaction semantics for reaction generation. However, ReGenNet directly models action-to-reaction generation process, without inferring action intention, thus achieving subpar performance.

To address these challenges, we propose Think-Then-React model (TTR), an LLM-based model designed to predict human reactions in online and unprompted settings with the following innovations: **First**, to unifiedly represent human motion in multi-person scenario, we propose decoupled space-pose tokenizers that separately handle egocentric pose features and absolute space features. Specifically, we train a VQ-VAE (Van Den Oord et al., 2017) to encode egocentric human pose sequences (i.e., the space features are normalized, to ensure codebook utilization) into LLM-readable tokens. To maintain spatial features which are crucial in multi-person interaction scenarios, we propose a space tokenizer that encodes positions and orientations as space tokens. We concatenate initial space tokens as prefixes to pose sequences, indicating the initial absolute state of an egocentric motion sequence. **Second**, to stabilize reaction prediction process, we introduce a novel framework that is capable to automatically infer text prompts for reaction generation. Specifically, TTR unifies two processes within one model: a **thinking** process that infers action intent and reasons reaction description, and a **reacting** process that takes both the action motion and inferred prompts as input, to generate precise and semantically appropriate reactions. **Third**, to adapt a language model to motion modality, we design a multi-task and multi-stage training pipeline consisting of motion-text, space-pose and motion-motion generation tasks. With our proposed training strategy, TTR is capable to effectively build correlations between text, motion and space modalities.

In summary, our main contributions are as follows:

- We introduce a unified motion tokenizer that effectively represents both absolute space and egocentric pose features into LLM-readable tokens in multi-person scenario.
- We propose a novel framework Think-Then-React with fine-grained training strategy, enabling the adaptation of a language model to a multi-modal model encompassing two processes: inferring action intention and reasoning reaction description, and predicting reaction, within one model, thus ensuring generation quality.
- Through extensive experiments, we demonstrate that our approach surpasses existing baselines by substantial margins, achieving an FID improvement from 3.988 to **1.942**, along with other ranking metrics.[1]

---

[1] Project page: `https://Think-Then-React.github.io/`.

# 2 RELATED WORK

## 2.1 HUMAN MOTION REPRESENTATION

Representing human motion can be mainly categorized into two norms: continuous and discrete representation. Human motion can be intuitively represented in continuous space as joint positions of 3D human skeleton extracted with SMPL (Loper et al., 2015). However, simply using joint position lacks enough information like joint velocity and rotation. Guo et al. (2022a) proposes redundant representation, consisting human root angular velocity, root linear velocity, root height, joint position, joint rotation, and foot-ground contact signals. This representation focuses on egocentric view in single-person scenario. Based on this, several works propose leveraging VQ-VAE (Van Den Oord et al., 2017) to encode human motion into discrete tokens, which can be fed into language models, adapting motion prediction task to language modeling task (Jiang et al., 2023; Guo et al., 2024; Zhang et al., 2023; Guo et al., 2022b). This technique is proved to be quite effective especially in the era of LLMs.

Representing human motion in multi-person scenario is more complicate than in single-person domain, as it is required to simultaneously representing egocentric pose and absolute space features (i.e., distance and orientation among multiple persons). Contrary to use normalized position and orientation in egocentric view, Liang et al. (2024) proposes a non-canonical representation that directly takes global signals (joint positions and velocities) as continuous motion representation, to maintain absolute information in multi-person scenario. Similar to Guo et al. (2022a), it combines joint position, velocity, rotation and foot-ground contact as continuous motion feature. Based on previous works, we propose a unified tokenizer, which decouples space and pose tokenization process, enabling effective adaptation of discrete motion representation to multi-person domain.

## 2.2 HUMAN MOTION GENERATION

The field of Human Motion Generation focuses on creating realistic and diverse 3D human motion from various input modalities, including text (Zhang et al., 2023; Guo et al., 2022b; Jiang et al., 2023; Guo et al., 2024; Liang et al., 2024), action labels (Guo et al., 2020; Xu et al., 2023; Petrovich et al., 2021a), and human motion (Chopin et al., 2023; Liang et al., 2024; Xu et al., 2024b). Most research has concentrated on text-conditioned single-person motion generation (text-to-motion) tasks. In this area, several works have utilized generative models commonly used in the vision domain, such as GANs, VAEs, and Diffusion Models, to generate human motion sequences. Another prominent approach (Zhang et al., 2023; Guo et al., 2022b; Jiang et al., 2023; Guo et al., 2024) employs VQ-VAE to encode human motion sequences into one-hot tokens, which are then processed by auto-regressive models. This method converts the high-dimensional generation task into a next-token prediction task, effectively leveraging pre-trained large language models for more accurate text prompt understanding and diverse motion generation.

Recently, there has been growing interest in generating human motion in multi-person scenarios. InterGen (Liang et al., 2024) introduces a dual-person interaction dataset with detailed textual descriptions and a diffusion-based model for jointly generating multi-person interactions conditioned on text input. InterFormer (Chopin et al., 2023) utilizes temporal and spatial attention with human skeleton priors to generate human motion sequences reacting to input action sequences. The latest work, ReGenNet (Xu et al., 2024b) employs a diffusion model to generate human reactions based on human actions in a unconstrained and online manner, and points out that given action's intention as a condition, the model can achieve superior performance compared to unconstrained settings. However, it directly predicts reaction motion without analyzing semantics of action motion. Our work unifies two processes: a thinking process that infers action semantics, and a reacting process that predicts reaction motion based on action motion and the thinking results, ensuring to generate reaction with appropriate semantics.

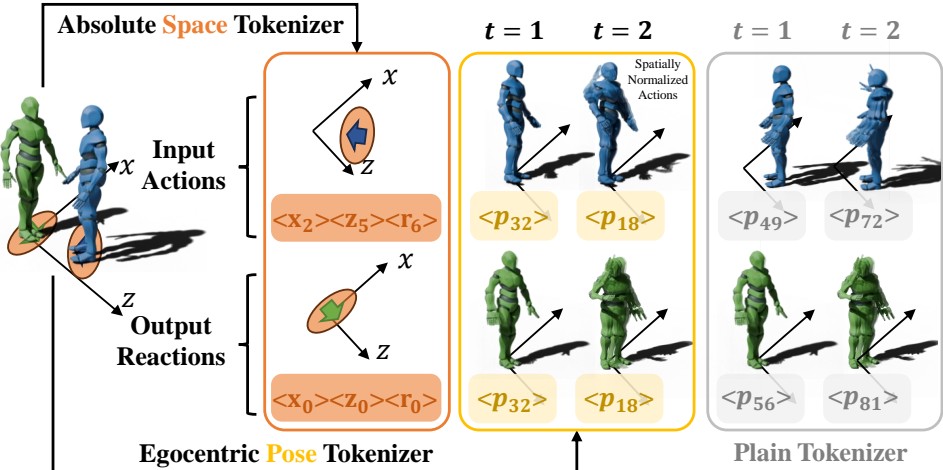

Figure 2: Illustration of our decoupled tokenizer and the plain tokenizer.

## 3 METHOD

### 3.1 OVERVIEW

For the task of action-to-reaction generation, given a human action $\mathbf{a} = \{a_i\}_{i=1}^{N_f}$ over $N_f$ frames, our aim is to generate a corresponding reaction $\mathbf{b} = \{b_i\}_{i=1}^{N_f}$ without any input prompts. Most previous works leverage input prompts but they are often inaccessible in unconstrained interaction setting. As the example in Figure 1 shows, when a robot/avatar meets a human, it can only observe the human behaviors, try to understand her/his intents, and think what the robot/avatar is expected to react. There is no prompt available to tell what they are going to do.

To address the above problem, we propose a unified framework, Think-Then-React (TTR), for both action understanding and reaction generation. First, we propose a unified tokenizer to convert both egocentric poses and absolute spatial location and orientation information into tokens (Section 3.2). Then, we propose a unified Large Language Model (LLM) based model that are pre-trained on three categories of motion and language related tasks, such as describing a motion, and then fine-tune the model with instructions of predicting a reaction from a given action (Section 3.3).

To avoid confusion, we define **pose** as a human body posture or movement within a brief time interval, such as "taking one step forward", and a pose can be represented as a single token. A **motion** refers to a sequence of poses, starting with an initial spatial state represented by space features. For example, a motion could be "a person walks three steps".

### 3.2 UNIFIED MOTION REPRESENTATION

To represent one or two persons (denoted by p1 and p2) in an absolute coordinate system, where the x-z plane represents the horizontal plane and y-axis represents the vertical direction, we normalize their centers at the origin while facing positive z axis. Then for each frame, we extract the 3D skeletons' joint position, velocity and rotation as normalized (or egocentric) pose feature. Before normalizing, we keep the two persons' pelvis 2D coordination $x$, $z$ and body orientation $r$ to maintain absolute space features. The y-axis (vertical) is not included, as few motions begin in a "floating" state. Based on pose and space features of p1 and p2, we propose a unified tokenizing pipeline to convert them into LLM-readable tokens.

### 3.2.1 EGOCENTRIC POSE TOKENIZER

Our aim is to convert continuous pose features into discrete pose tokens like "$<p_{128}><p_{42}>...$". To achieve this, we adopt VQ-VAE (Van Den Oord et al., 2017), similar to Jiang et al. (2023), as the egocentric pose tokenizer. The pose tokenizer consists of an encoder $\mathcal{E}$ and a decoder $\mathcal{D}$. $\mathcal{E}$ first encodes continuous motion features, i.e., the 22 joints' position, rotation and velocity vector

$\mathbf{m} = \{m_i\}_{i=1}^{N_f}$ into $N_t$ discrete pose tokens, i.e., $N_t$ timesteps, which is downsampled from $N_f$. Specifically, $\mathcal{E}$ and $\mathcal{D}$ are 1D convolution networks with downsample and upsample blocks.

We first obtain the latent pose representation of a motion sequence $\hat{\mathbf{p}} = \mathcal{E}(\mathbf{m})$. Then, we set up a learnable codebook for human poses $P \in \mathbb{R}^{N_p \times d_p}$ with $N_p$ entries in size $d_p$. A quantization operation $Q(\cdot)$ is applied on the encoded motion latent features by replacing each row vector $\hat{\mathbf{p}}_i \in \hat{\mathbf{p}}$ with its nearest codebook entry $\mathbf{p}_k$. The process is formulated as:

$$\mathbf{p}_{quantized} = Q(\hat{\mathbf{p}}) := (\arg\min_{\mathbf{p}_k \in C} ||\hat{\mathbf{p}}_i - \mathbf{p}_k||) \in \mathbb{R}^{N_p \times d_p} \tag{1}$$

Then, we obtain the reconstructed pose feature $\hat{\mathbf{m}}$ through the decoder $\hat{\mathbf{m}} = \mathcal{D}(\mathbf{p}_q)$. The overall process of the VQ-VAE can be formulated as:

$$\hat{\mathbf{m}} = \mathcal{D}(Q(\mathcal{E}(\mathbf{m}))). \tag{2}$$

This is trained via a reconstruction loss with codebook commitment loss. Noting that the $argmin$ operation is non-differentiable, we simply copy the gradients from $\mathcal{D}$ to $\mathcal{E}$ as the estimated gradient. Furthermore, for smoother reconstructed motion and a stable training process, we add an extra velocity regularization in the reconstruction loss and employ exponential moving average (EMA) Hunter (1986) with codebook reset techniques, following Zhang et al. (2023). More details about this section are provided in Section A.1.

### 3.2.2 ABSOLUTE SPACE TOKENIZER

For better generalization capability, all motions, including actions and reactions, are normalized to the original point and same direction before being tokenized. Therefore, absolute space information, i.e., the human body 2D position and orientation of each person, is omitted. To extend egocentric pose tokens with absolute space information, we propose converting position and rotation of a person's center point into LLM-readable tokens.

As shown in Figure 2, before normalizing a human motion, we first extract the center point's features, i.e., the position $x$ and $z$ and orientation $r$. We then compute the range of $x$, $z$, and $r$ across the dataset to get the maximum and minimum values. These ranges are uniformly divided into $N_b$ bins, converting each continuous value to discrete tokens. For example, $x = 0.55$ will be represented as "$<x_{15}>$" if all the x positions are in range $[-1, 1]$ with $N_b = 20$ bins.

Finally, we use a unified coding system to represent action, reaction, and their relative information. Specifically, for each motion, we apply absolute space tokenizer to encode initial $x$, $z$, and $r$ into egocentric pose tokens, and apply pose tokenizer to encode the following pose sequence, i.e., the following motion, into pose tokens. Such tokens enable training a model that can understand and generate motion and language simultaneously effectively and efficiently in the subsequent phase.

### 3.3 UNIFIED LLM BASED MOTION UNDERSTANDING AND GENERATION

#### 3.3.1 PRE-TRAINING

To adapt a language model into a motion-language model, we first pre-train the model with multiple tasks in diverse formats. The pre-training tasks can be categorized into three main types:

**(1) Motion - Text.** To enable the model to understand and generate human motion, we combine the action and reaction token sequences to construct prompts, which are then fed into the model to generate corresponding textual descriptions, and vice versa. For example, the input sequence could be "Describe the interaction. Action: $<x_0><z_1><r_2><p_2><p_7>$..., Reaction: $<x_7><z_7><r_8><p_1><p_9>$...", and the target response is: "*One waves the right hand, and the other one waves back*". However, reaction motions are not given during the inference phase. Therefore, the reaction motion is randomly dropped during the training phase to enable the model to infer the interaction from the action motion solely. We also pre-train our model by the instructions on predicting action and reaction token sequences from an interaction description prompt in text.

**(2) Pose - Space.** Spatial information is represented by orthogonal one-hot tokens, but it may be helpful to infuse auxiliary spatial information into the model. Specifically, we design two tasks: i) Egocentric pose to absolute space: Given space token and subsequent pose tokens of $t$ timestep, we

train the model to predict the space tokens of $t + 1$ timestep. For example, given input space token $<z_{12}>$ and a pose token $<p_{56}>$, which represents "stepping forward", the target output should be $<z_{13}>$, denoting spatial transition. ii) Absolute space to egocentric pose: Similarly, given space tokens of $t$ and $t + 1$ timestep, the model is trained to predict pose tokens between them.

**(3) Motion - Motion.** To capture more fine-grained action-reaction relationships, we use the first half of the action sequence and the second half of the reaction sequence, along with their corresponding initial spatial tokens, as input. The model is then pre-trained to complete the remaining motion clips. For example, given a sequence spanning ten timesteps $t_{1:10}$, we feed the first half of the action $a_{1:5}$ and the second half of the reaction $b_{6:10}$, supervising the model to predict $a_{6:10}$ and $b_{1:5}$. Alternatively, we feed $b_{1:5}$ and $a_{6:10}$ to predict $a_{1:5}$ and $b_{6:10}$.

During pre-training, we jointly train all the tasks in a non-causal manner for better efficiency. Owing to our unified motion and language architecture and space-pose token representation, single person motion and text data can be seamlessly integrated into the training process. We adopt HumanML3D (Guo et al., 2022a), a large scale single person motion-text dataset to facilitate pre-training. To avoid overfitting, we prepare 20 prompt templates for each task and randomly mask out 15% of tokens to be predicted during training. In addition, we adopt random clipping of motions as augmentation. We also find that text generation tasks converge much faster than motion generation tasks. To balance different training tasks, we use the validation losses of the tasks as sampling weights to dynamically select the training source for each epoch.

### 3.3.2 FINE-TUNING

After pre-training, the motion-language model is well-structured with knowledge of pose, space, and text. To make the model applicable to online action-to-reaction generation, we fine-tune it in a causal manner, focusing on two tasks: thinking and reacting.

The **thinking** task involves understanding action motion, e.g., "the person is waving hand", and inferring its possible interaction, e.g., " two persons wave goodbye to each other", or reaction, "the other person waves back". At each training iteration, we randomly choose the first quarter, half, or the entire action sequence as input to predict the entire interaction caption. However, the entire action motion is not given in the early stage of inference, thus the inferred description based on action motion clips may not be accurate, thus we adopt periodical **re-thinking** in the inference phase for each $N_r$ action tokens given, to dynamically adjust the prompt for reaction generation. We define $N_r$ as re-thinking interval.

For the **reacting** task, we aim to supervise the model to generate reaction motions conditioned on the generated descriptions during the thinking process. However, in the early stages of fine-tuning, the inferred interaction descriptions are not accurate enough to guide the reaction generation process. Thus, we adopt a teacher forcing approach. In the early stages, the model takes the ground-truth text prompt as a condition to generate the entire reaction sequence. Meanwhile, we monitor the validation loss and text generation metrics. When the metrics tend to converge, we begin to sample predicted interaction captions by the model and use them as reaction generation conditions. This process ensures alignment between training and inference, as ground-truth prompts are inaccessible during inference.

## 4 EXPERIMENT

We evaluate our proposed method with strong baselines and further analyze contributions of different components, and the impact of key parameters.

### 4.1 EXPERIMENT SETUP

**Dataset.** We evaluate all the methods on Inter-X dataset, which consists about 9K training samples and 1,708 test samples. Each sample is an action-reaction sequence and three corresponding textual description. As supplementation, we mix our pre-training data with single person motion-text dataset HumanML3D (Guo et al., 2022a), which consists more than 23K annotated motion sequences. We uniformly sample frames for both datasets to 30 FPS.

**Evaluation Metrics.** Following single-person motion generation (Zhang et al., 2023), we adopt the these metrics to quantitatively evaluate the generated motion: R-Precision measures the ranking of Euclidean distances between motion and text features. Accuracy (Acc.) assesses how likely a generated motion could be successfully recognized as its interaction label, like "high-five". Frechet Inception Distance (Heusel et al., 2017) (FID) evaluates the similarity in feature space between predicted and ground-truth motion. Multimodal Distance (MMDist.) calculates the average Euclidean distance between generated motion and the corresponding text description. Diversity (Div.) measures the feature diversity within generated motions. All the metrics reported are calculated with batch size set to 32, and accumulated across the test dataset, and we evaluate each method for 20 times with different seeds to calculate the final results at 95% confidence interval.

**Evaluation Model.** Every metric mentioned above requires an encoder $\mathcal{M}$ to extract motion feature. For single person text-to-motion generation tasks, a motion-text matching model are commonly trained as human motion feature extractor. A simple way to transfer this method to interaction domain is to directly train an interaction-to-text matching model $\mathcal{M}(\mathbf{a}, \hat{\mathbf{b}}, text)$, where action sequence $\mathbf{a}$ and predicted reaction sequence $\hat{\mathbf{b}}$ together is regarded as a generated interaction sequence, or a reaction-to-text match model $\mathcal{M}(\hat{\mathbf{b}}, text)$. However, the former one may focus too much on the ground-truth action input, leading insufficient discriminative power of $\hat{\mathbf{b}}$'s quality, while the latter one lacks semantics provided by action, thus leading to subpar matching capability.

To address the issue, we simply uniformly mask off a large portion of $\mathbf{a}$, obtaining down-sampled action motion sequence $\mathbf{a}'$ (downsampled to 1 FPS in our setting), which serves as a semantic hint for the matching process while not introducing too much emphasis on input action sequence. The final evaluation model consists of an masked interaction encoder and a text encoder. We use contrastive loss following CLIP (Radford et al., 2021), which encourages paired motion and text features to be close geometrically. In addition, we add a classification head after the predicted motion features, to simultaneously predict interaction labels, such as "high-five".

**Baselines.** To evaluate the performance of our method TTR on online and unconstrained setting, we compare TTR with the following baselines: 1) **InterFormer** (Chopin et al., 2023) is a transformer based action-to-reaction generation model that leverages human skeleton as prior knowledge for efficient attention process. 2) **MotionGPT** (Jiang et al., 2023) is a motion-language model that leverages an LLM for motion and text generation. We extend the motion tokenizer of MotionGPT to encode multi-person motion, while keeping other settings unchanged. 3) **InterGen** (Liang et al., 2024) proposes a mutual attention mechanism within diffusion process for human interaction generation, we reproduce and adapt IngerGen to action-to-reaction generation. 4) **ReGenNet** (Xu et al., 2024b) is latest state-of-the-art model on action-to-reaction generation. It adopts a transformer decoder based diffusion model, which directly predicts human reaction given action input in unconstrained and online manner as ours.

**Implementation Details.** For the LLM, we adopt Flan-T5-base (Chung et al., 2024; Raffel et al., 2020) as our base model, with extended vocabulary. We warm up the learning rate for 1,000 steps, peaking at 1e-4 for the pre-training phase, and use the same learning rate for fine-tuning. Both the pre-training and fine-tuning phases are trained on a single machine with 8 Tesla V100 GPUs. The training batch size is set to 32 for the LLM and we monitor the validation loss and reaction generation metrics for early-stopping, resulting about 100K pre-training steps and 40K fine-tuning steps. We set the re-thinking interval $N_r$ to 4 tokens and divide each space signal into $N_b = 10$ bins.

## 4.2 COMPARISON TO BASELINES

As shown in the upper side of Table 1, our method TTR significantly outperforms baseline methods in terms of ranking, accuracy, FID and multimodal distance, showing superior human reaction generation quality. Compared to MotionGPT, which adopts a similar motion-language architecture, TTR expresses stronger performance, which we attribute to our unified representation of motion via space and pose tokenizers, enabling effective individual pose and inter-person spatial relationship representation. TTR also surpasses the diffusion-based methods, InterGen and ReGenNet, with our think-then-react architecture, improving generated motions by describing observed action and reasoning what reaction is expected on semantic level. In addition, ReGenNet and MotionGPT get closer diversity to the real than our model. We mainly attribute to that, TTR may conduct multiple re-thinking processes during inference, and the inferred semantics may bring a higher diversity.

Table 1: Comparison to state-of-the-art baselines and ablation studies of our method on Inter-X dataset. ↑ or ↓ denotes a higher or lower value is better, and → means that the value closer to real is better. We use ± to represent 95% confidence interval and highlight the best results in **bold**. For ablation methods (in grey), PT, M, P, S, and SP are abbreviations for pre-training, motion, pose, space, and single-person data, respectively.

| Methods | R-Precision↑ Top-1 | Top-2 | Top-3 | Acc.↑ | FID↓ | MMDist↓ | Div.→ |
|---|---|---|---|---|---|---|---|
| Real | $0.511^{\pm.003}$ | $0.682^{\pm.002}$ | $0.776^{\pm.002}$ | $0.463^{\pm.000}$ | $0.000^{\pm.000}$ | $5.348^{\pm.002}$ | $2.498^{\pm.005}$ |
| InterFormer | $0.172^{\pm.012}$ | $0.292^{\pm.013}$ | $0.343^{\pm.012}$ | $0.171^{\pm.009}$ | $10.468^{\pm.021}$ | $7.831^{\pm.018}$ | $3.505^{\pm.023}$ |
| MotionGPT | $0.238^{\pm.003}$ | $0.354^{\pm.004}$ | $0.441^{\pm.003}$ | $0.186^{\pm.002}$ | $5.823^{\pm.048}$ | $6.211^{\pm.005}$ | $2.615^{\pm.007}$ |
| InterGen | $0.326^{\pm.036}$ | $0.423^{\pm.063}$ | $0.525^{\pm.053}$ | $0.254^{\pm.019}$ | $5.506^{\pm.257}$ | $6.182^{\pm.038}$ | $2.284^{\pm.009}$ |
| ReGenNet | $0.384^{\pm.005}$ | $0.483^{\pm.002}$ | $0.572^{\pm.003}$ | $0.297^{\pm.004}$ | $3.988^{\pm.048}$ | $5.867^{\pm.009}$ | $\mathbf{2.502^{\pm.001}}$ |
| TTR (Ours) | $\mathbf{0.423^{\pm.005}}$ | $\mathbf{0.599^{\pm.003}}$ | $\mathbf{0.693^{\pm.003}}$ | $\mathbf{0.318^{\pm.003}}$ | $\mathbf{1.942^{\pm.017}}$ | $5.643^{\pm.003}$ | $2.629^{\pm.006}$ |
| w/o Think | $0.367^{\pm.003}$ | $0.491^{\pm.027}$ | $0.584^{\pm.008}$ | $0.230^{\pm.036}$ | $3.828^{\pm.016}$ | $6.186^{\pm.055}$ | $2.609^{\pm.006}$ |
| w/o All PT. | $0.398^{\pm.007}$ | $0.531^{\pm.002}$ | $0.628^{\pm.003}$ | $0.288^{\pm.002}$ | $3.467^{\pm.113}$ | $5.822^{\pm.003}$ | $2.909^{\pm.053}$ |
| w/o M-M PT. | $0.408^{\pm.005}$ | $0.563^{\pm.004}$ | $0.646^{\pm.005}$ | $0.293^{\pm.004}$ | $2.874^{\pm.020}$ | $5.736^{\pm.003}$ | $2.553^{\pm.006}$ |
| w/o P-S PT. | $0.417^{\pm.004}$ | $0.582^{\pm.004}$ | $0.664^{\pm.004}$ | $0.308^{\pm.003}$ | $2.685^{\pm.024}$ | $5.699^{\pm.004}$ | $2.859^{\pm.007}$ |
| w/o M-T PT. | $0.406^{\pm.003}$ | $0.557^{\pm.004}$ | $0.637^{\pm.004}$ | $0.304^{\pm.003}$ | $2.580^{\pm.021}$ | $5.822^{\pm.003}$ | $2.889^{\pm.005}$ |
| w/o SP Data | $0.414^{\pm.004}$ | $0.592^{\pm.005}$ | $0.685^{\pm.003}$ | $0.315^{\pm.004}$ | $2.007^{\pm.015}$ | $5.667^{\pm.003}$ | $2.611^{\pm.005}$ |

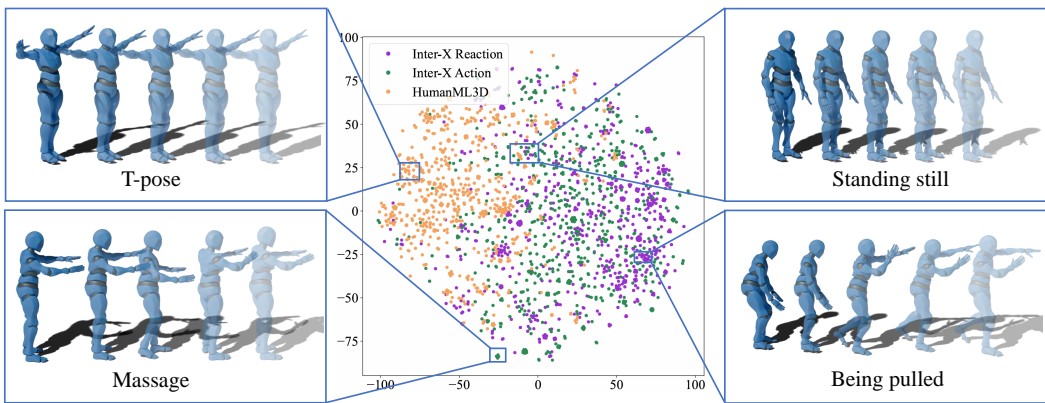

Figure 3: Visualization of a person's motion sequences in Inter-X dataset and HumanML3D dataset.

## 4.3 ABLATION STUDY OF KEY COMPONENTS

To evaluate the effectiveness of our proposed key designs, we conduct detailed ablation studies by removing each of them to observe how much drop compared to the full version of our TTR method. The larger drop indicates more contribution. The results are shown in gray lines of Table 1. According to the drops in FID, all designs, including thinking, pre-training tasks and using single person data in pre-training, have positive contributions to the final performance, and thinking contributes the most. Some detailed findings and analyses are as follows.

First, we skip **thinking** stage during inference, and find the performance drops significantly in FID from 1.9 to 3.8. This supports the necessity of our proposed thinking process before reacting. We also notice decreasing diversity of generated samples, as the model relies solely on input action, and cannot explicitly capture and infer action's intent, thus leading to more rigid motion in some cases.

Second, to evaluate the effectiveness of **pre-training**, we omit the pre-training stage, and directly train our model TTR for thinking and reacting tasks. As shown in Table 1, our model's performance deteriorates without a fine-grained pre-training phase from 1.9 to 3.4 in FID. This indicates that pre-training can effectively adapt a language model (Flan-T5-base) into a motion and language model. We further removing three kinds of pre-training tasks: motion-motion (M-M PT.), pose-space (P-S PT.), and motion-text (M-T PT.). The results show that the without any task, the performance obviously gets worse, from 1.9 to 2.5 - 2.8 in FID, indicating their positive contribution to the final performance and complementary values to each other.

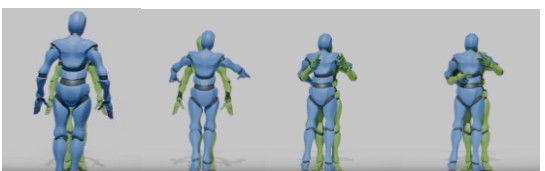

(a) Two people stand facing each other. One person approaches and opens her/his arms to embrace the other person's back and waist, while the other person imitates the same action.

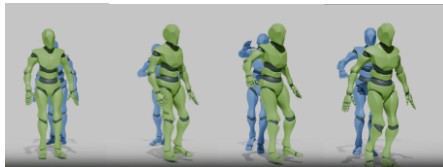

(b) The first person pushes the second person heavily on the back with both hands, causing her/him to be pushed forward several steps.

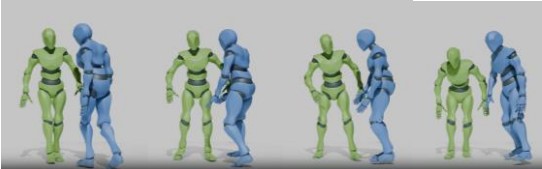

(c) The first person runs towards the other and knocks her/his left shoulder against the right shoulder, and the second person is forced to step back.

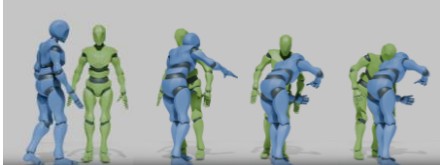

(d) The first person grabs the other person's waist, the second person wrestles with the first person.

Figure 4: Visualized cases of our predicted reactions (in green) to input action (in blue) and corresponding thinking results. We also provide a failure case in figure (d), where TTR misunderstands the input action as "wrestling", which should be "embracing".

Third, to see how much **single-person data** helps reaction generation, we remove single person motion-text data, i.e., the data from HumanML3D dataset, from our training set. The result (w/o SP Data) shows that the model performs worse without training on HumanML3D, which proves that our unified motion encoder and motion-language architecture can leverage both single- and multi-person data, alleviating the insufficiency of training data. However, the benefit from single-person data is not as large as we expect.

## 4.4 ANALYSIS ON OVERLAPPING BETWEEN SINGLE- AND MULTI-PERSON MOTIONS

To investigate the reason of small contribution from single-person data, we further visualize motion sequences of single-person motion (HumanML3D), two-person action (Inter-X Action) and reaction (Inter-X Reaction) in the same space, as presented in Figure 3. Specifically, we use t-SNE tool Van der Maaten & Hinton (2008) to project motion token sequence features into two-dimension. As shown in Figure 3, the single- and two-person motion sequences have little overlap. When doing case studies, we find that most two-person motion are unique, e.g., massage and being pulled, and will never be used in single-person motion. Similarly, most single-person motions are unique too, e.g., T-pose, and seldom appear in multi-person interaction. There are only a few overlapped motions, e.g., standing still. In addition, when comparing action and reaction sequences in multi-person interaction, we have some interesting findings. When reactions are close to actions, the motion usually belongs to symmetrical interactions, e.g., pulling or being pulled; whereas, when actions are far from reactions, the motion usually belongs to asymmetrical interaction, e.g., massage.

## 4.5 IMPACT OF DOWN-SAMPLING PARAMETER IN MATCHING MODEL FOR EVALUATION

As described in Section 4.1, we propose downsampling action motion sequence to avoid matching models for evaluation pay too much attention to input action rather than output reaction. We conduct an experiment to change the downsampling parameter frame rate and calculate the difference between taking ground-truth action and random action as the input of $\mathcal{M}$, in terms of summed ranking scores (Top-1, Top-2, Top-3 and Acc.). As presented in Figure 5, difference is lowest when FPS equals to 0, which meaning we only match generated reaction motion with text. It goes up to the peak when FPS equals 1 and quickly goes down to low values, even close to the lowest when FPS is about 15. This indicates that it is necessary to concatenate input action with generated reaction to compose a meaningful interaction in evaluation, otherwise the motion-text matching model cannot well recognize the interaction. However, only 1 FPS is enough. With larger FPS, the matching

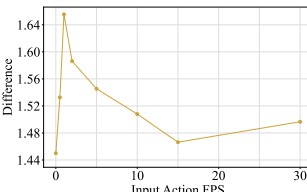

Figure 5: Impact of input action FPS to summed ranking score differences.

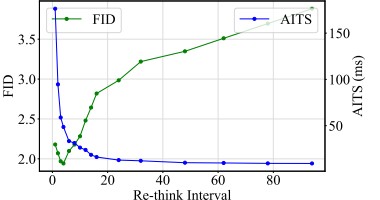

Figure 6: Impact of re-thinking interval to FID and average inference time per step (AITS).

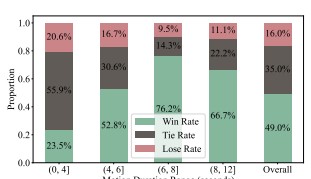

Figure 7: User preference between TTR and ReGenNet on different motion duration.

models will be disturbed by input action rather than the generated reaction. Thus, we choose 1 FPS, corresponding to the largest difference, as our final setting.

### 4.6 IMPACT OF RE-THINKING INTERVAL

We change the re-thinking interval $N_r$ from about 1 to 100 timesteps (about 0.1 to 10 seconds) and observe how it impacts generative quality measure FID. As shown in Figure 6, FID falls down first until $N_r = 4$ (about 0.5 second) and then continues rising up. This indicate that the best time interval is about 0.5 second. When the time interval is too short, our TTR model cannot get enough information to re-think what the input action means and will bring some randomness into predicting appropriate reaction. When the time interval gets too long, our TTR model give slow responses to the input action sequences and generates coarse-grained reaction.

We also evaluate the average inference time per step (AITS) with respect to the re-thinking interval. As shown in Figure 6, the inference time significantly decreases as the re-thinking interval increases, eventually converging to approximately 10 milliseconds per step (100 FPS). In our setup, we opt to re-think every four steps, resulting in an inference time of less than 50 milliseconds, which meets the requirements for a real-time system.

### 4.7 USER STUDY

To further evaluate our model qualitatively, we conduct a user study on TTR vs. the latest SOTA method ReGenNet, and the results are shown in Figure 7. We randomly sample 100 action sequences from Inter-X dataset, which are fed into TTR and ReGenNet to predict reactions, and ask four real human to choose the better ones. It can be seen that TTR surpasses ReGenNet on all the duration range, and the winning rate rises significantly when motion duration is longer. We mainly contribute this to our explicit thinking and re-thinking procedure, which ensures semantics matching and alleviates accumulated errors.

## 5 CONCLUSION

In this paper, we propose a novel framework Think-Then-React (TTR) to address the action-to-reaction motion generation problem. First, we propose a unified motion encoder that tokenizes a person's starting location and following poses separately. Then we design motion and text related tasks to pre-train a large language model backbone to understand and generate both language and motion. We also fine-tune the model to think what the input action means and what an appropriate reaction is, and then generate reaction motions. Experimental results show that our proposed TTR method outperforms all baselines in all metrics except for diversity. Our proposed thinking phase and all pre-training tasks contribute to the best performance. We find that although our proposed unified motion encoder enable leveraging single-person data in pre-training, it brings limited benefit due to the little overlapped poses between single-person motion and multi-person interaction. In the future, we plan to explore more effective method for single-person and multi-person dataset.

**Acknowledgment** This work is supported by the National Natural Science Foundation of China (No. 62276268) and Kuaishou Technology.

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

## A APPENDIX

### A.1 MOTION REPRESENTATION AND POSE TOKENIZER

For motion representation, we use the same strategy asLiang et al. (2024), which combines local joint positions, rotations, velocities, and foot-ground contact as the feature of human motion. Regarding the tokenizers, we adopt a temporal down-sample rate of four and $N_p = 256$ for motion tokens, each motion token are in $d_p = 512$ in the codebook. We divide all space tokens into $N_b = 10$ bins. The motion VQ-VAE is trained for 150K steps with batch size set to 256 and learning rate fixed at 1e-4 on a single Tesla V100 GPU. We adopt a similar architecture to Guo et al. (2022a) as our pose tokenizer. The encoder/decoder consists of two down-sample/up-sample 1D convolution layers and three 1D ResNet blocks He et al. (2016). We set the width of the auto-encoder to 512. We train the model on both the Inter-X and HumanML3D datasets for 200,000 steps, with batch size set to 256, and learning rate set to 1e-4. We apply L1-loss on both pose feature and velocity reconstruction, and a commitment loss for the embedding process. The weight set to velocity loss is 0.5 and commitment loss is 0.02.

### A.2 MATCHING MODEL

For the motion-text matching model, we adopt a similar architecture to InterCLIP (Liang et al., 2024), which consists of an eight-layer motion transformer encoder and an eight-layer text transformer encoder. The hidden size is set to 768 and attention heads is set to 8. We add a learnable token to the motion encoder and extract its feature in the last layer of motion encoder as the pooled motion feature. To perform motion classification, we add a classification head (an MLP) after the pooled motion feature. We use the text embedding layer from clip-vit-large-patch14 (Radford et al., 2021), which is frozen during training. We train the model for 40 epochs with batch size set to 128. The learning rate is warmed-up to 0.001 in the first 1,000 steps.

### A.3 EVALUATION ON MOTION CAPTIONING TASK

Table 2: Motion captioning results on Inter-X dataset. TTR* denotes feeding both action and reaction motion into TTR for captioning. TTR ($x$%) denotes only the first $x$% of action motion is fed into TTR for captioning.

|        | RAEs | SeqGan | Seq2Seq | TM2T | TTR* | TTR | TTR (50%) | TTR (25%) |
|--------|------|--------|---------|------|------|------|-----------|-----------|
| Bleu-1 | 28.6 | 45.4   | 53.8    | 56.8 | **60.2** | 55.6 | 54.1 | 52.2 |
| Bleu-4 | 9.7  | 14.1   | 18.5    | 21.6 | **25.4** | 20.3 | 18.9 | 16.6 |
| Rouge  | 34.1 | 36.8   | 45.2    | 48.2 | **50.5** | 46.4 | 45.3 | 43.0 |

We also evaluate our TTR model on motion captioning task, and the results are shown in Figure 3. The results of baselines are from Inter-X paper Section A.1. As the baseline methods all take both action and reaction as input, while in our setting, our thinking process is only accessible to ground-truth action, we first align TTR's setting with the baselines', denoted as TTR*. It can be seen that, with our fine-grained training and effective motion representation, TTR* achieves the best captioning performance in all metrics.

Then we evaluate TTR on real-world settings, i.e., only partial of the input action is visible to our model. We take the first 25%, 50% and full action as input of TTR for the action-to-text generation process. It can be seen that even though only a quarter of input action is given, TTR is still capable of accurately predicting the corresponding action and reaction description, showcasing strong generalization capability.

### A.4 ABLATION STUDY ON THINKING PROCESS

To evaluate the necessity of the Thinking process, we conduct an ablation study on different prompts provided to the Reacting process. First we fed ground-truth prompt to the Thinking process, and it can be seen that the overall quality of predicted reaction is significantly improved. Then we

Table 3: Ablation study on how does thinking process influence model performance. GT denotes ground-truth, and Thinking$^*$ denotes using a better motion-to-text model for the thinking process.

| Methods | FID | Top-1 | Acc. |
|---|---|---|---|
| w/ GT Prompt | $1.584^{\pm.016}$ | $0.458^{\pm.005}$ | $0.361^{\pm.005}$ |
| w/ Thinking$^*$ | $1.882^{\pm.014}$ | $0.429^{\pm.004}$ | $0.331^{\pm.003}$ |
| w/ Thinking | $1.942^{\pm.017}$ | $0.423^{\pm.005}$ | $0.318^{\pm.003}$ |
| w/o Thinking | $3.828^{\pm.016}$ | $0.367^{\pm.003}$ | $0.230^{\pm.036}$ |

leverage a enhanced Thinking model as mentioned in Section A.3, and the FID decreases from 1.94 to 1.88, proving that a better thinking process leads could promote the following Reacting process. Moreover, when discarding the Thinking process, our model dramatically deteriorates in reaction generation quality, as Thinking and re-thinking process is crucial to guide reaction generation and reduce accumulated errors.

