# OpenReview forum: "Think Then React: Towards Unconstrained Action-to-Reaction Motion Generation"
_ICLR.cc/2025/Conference — ICLR 2025 Poster_

### Official Review · Reviewer_wKay · 2024-10-24

**Soundness:** 3
**Presentation:** 3
**Contribution:** 3
**Rating:** 8
**Confidence:** 4

**Summary:**

The paper present a new method for generating a reaction motion based on an action motion in a two persons interaction. The proposed architecture uses LLMs to understand the input action motion and then infer possible reactions to this action. The action motion is first converted into tokens that are used by the LLM to infer the type of action and predicts the reaction and the corresponding token that are then projected back into motion space. The method uses several training schemes to achieve a better understanding of the action and generate a better reaction. The method outperform the state of the art quantitatively on the Inter-X dataset.

**Strengths:**

- The idea of splitting the task of reaction generation between a "thinking" process using LLM and a "reacting" process using VQ-VAE is very interesting. This method will probably be able to scale well as new human interaction datasets are released.
- The architecture and training scheme as well as the intuitions behind them are sound.
- Treating the global pose with a separate tokenizer is a good idea and allow for more flexibility while staying accurate.
- "re-thinking" is also interesting to correct the generation in an online manner.
- Method beats the state of the art by a good margin on most metrics.
- Extensive ablation on the pretraining.

**Weaknesses:**

- Very few qualitative results. There is no qualitative comparison against the state of the art and no videos to show the actual motions. the lack of qualitative results makes it very difficult to evaluate the model qualitatively and to see if quantitative improvements correspond to similar qualitative improvements.
- The "pose tokenizer" described in the main paper (3.2.1) is different from the one described in the appendix. Is the tokenizer from the appendix the "space tokenizer" ? This should be clarified.
- Sec 3.3.1 (3) "we take the first half of action and the second half of reaction sequence of tokens as input". Is the opposite operation also performed ? i.e. "take the SECOND half of action and the FIRST half of reaction sequence of tokens as input" ? The text says "and vice versa" but it is not clear what it refers to.
- The authors do not explain why they use only the 2D positions for the "absolute pose" and not the 3D positions.
- There are some issues with notations:
    - line 204 "k" is never defined and why take "k-l" as input ?
    - line 231 "M" is never defined.
    - line 231 why change the notation from "Nf" to "L" for the number of frame ?
    - line 266 why use "t" for timestep when it was defined as "l" (or maybe "k") on line 204 ?
- Some of the writing could use improvement:
    - lines 203-206 : phrasing is very confusing and makes it difficult to understand what the authors want to say.
    - lines 266-268 : same as above
    - The way "pose" and "motion" have been defined and are then used is sometime confusing.

**Questions:**

- Qualitative comparison are very important in these type of works. Including them is always a good idea as they help in showing the performance of the method. Also since the work is on motion video qualitative results are always appreciated.
- Typo line 231 : "number of frames or original motion"

---

> ### Author Response · Authors · 2024-11-22
> **Response to Reviewer wKay**
>
> We deeply appreciate your positive evaluation and encouraging feedback on our work. Below, we address your questions and suggestions in detail.
>
> - **Q1. Qualitative results and comparisons**
>
> Good remark! We have provided qualitative comparisons and failure cases on an anonymous webpage: Think-Then-React.github.io. We have included some of them in our revised paper (Line 432-449).
>
> - **Q2. Pose tokenizer in main paper and appendix**
>
> We introduce the **architecture** of our pose tokenizer in main paper Section 3.2.1 and elaborate **implementation details** of the pose tokenizer in Appendix A.1, rather than space tokenizer. We notice that the citations between these sections are inconsistent, as both of the related works use a similar VQ-VAE architecture. We have now corrected these inconsistencies between the main paper and the appendix (Line 652).
>
> To further clarify, the **space tokenizer** is responsible for encoding spatial features, such as the (x, z) position and human body orientation (e.g., <x0, z0, r0> denotes a person facing the positive z-direction at the center point). This maintains absolute spatial information for both individuals. The **pose tokenizer** auto-encodes normalized human motions into token sequences (e.g., <p32><p48><p96> could describe "one person reaching out with their right hand"), which can then be used by LLMs for understanding and predicting reactions.
>
> If there are any further inconsistencies or points that need clarification, please let us know, and we would be happy to address them.
>
> - **Q3. Pose-to-Pose training detail**
>
> The phrase "and vice versa" refers to the fact that we perform both operations: 1) Predicting a_second and b_first from a_first and b_second, and 2) Predicting a_first and b_second from a_second and b_first. We have clarified this in Section 3.3.1 (Line 289-291).
>
> - **Q4. Exclusion of y-axis from space representation**
>
> Most motions typically begin with the y-coordinate set to 0 (as they generally don’t start in mid-air), so we exclude the y-axis from the absolute pose for simplicity and to avoid introducing redundant information. However, subsequent pose tokens still capture vertical motion dynamics, such as jumping or climbing, through the changes in the encoded token sequence. We have added relevant clarification in Section 3.2 (Lines 217-218).
>
> - **Q5. Notation and phrasing issues**
>
> We have revised inconsistent notations and improved clarity in Sections 3 and 4, including corrections for "k," "M," and "L" and unifying timestep notation with t. Additionally, we rephrased unclear passages in Sections 3.3.1 and 4.
>
> - **Q6: Definition between pose and motion**
>
> To clarify the terminology, we use following definitions: 1) **Pose** refers to a human body posture or movement within a brief time frame (i.e., timestep), such as "taking one step forward." A pose can be represented as a single token. 2) **Motion** is a sequence of poses, starting with an initial spatial state represented by space features. For example, a motion might be "a person walks three steps."

---

> > ### Comment · Reviewer_wKay · 2024-11-25
> > **review after rebuttal**
> >
> > I thanks the authors for their extensive rebuttal. After reviewing it, I find that the authors answered all of my concerns. I see no reason to change my rating.

---

> > > ### Author Response · Authors · 2024-11-25
> > >
> > > We are glad that we could address these concerns, and we sincerely appreciate your recognition!

---

### Official Review · Reviewer_enFt · 2024-11-03

**Soundness:** 3
**Presentation:** 3
**Contribution:** 2
**Rating:** 6
**Confidence:** 4

**Summary:**

The paper presents Think-Then-React, a novel method for action-to-reaction motion generation. It introduces a unified motion encoder that tokenizes the initial position and subsequent poses. Besides it employs a large language model for understanding and generating both motion and text. The model is pre-trained on various tasks, therefore it enhances its performance in generating coherent reactions based on observed actions.

**Strengths:**

- The paper aims at improve the quality of action-to-reaction generation, which is an important issue.
- The paper introduces a unified space-pose token representation and proves its effectiveness.
- The significant improvement in performance compared to the baseline is a major advantage.

**Weaknesses:**

- The training data utilized a mixture of two datasets, which may lead to an unfair comparison.
- The experiments only presented numerical metrics on the test set and did not include evaluations of the generated actions by real humans.

**Questions:**

- I am confused by the dataset used for training. Do other methods also use mixed dataset? If they only use one dataset but you use the mixture of two, it will cause unfairness. Please provide the information about the dataset used by other methods in the paper and the details about dataset mixture strategies.

- Does your paper include a comparison involving evaluations of the generated actions by real humans (i.e., a user study)? In the generation field, the importance of user study results is comparable to that of numerical metrics.

---

> ### Author Response · Authors · 2024-11-22
> **Response to Reviewer enFt (1/2)**
>
> We are grateful for your thoughtful suggestions and recognition of our contributions. Please find our detailed responses to your concerns below.
>
> - **Q1. Mixture of two datasets**
>
> Baseline methods do not use mixture datasets, as they typically cannot leverage single-person datasets like HumanML3D without a unified representation, which encodes both single- and multi-person scenarios. The reason we use mixture dataset is because we aim to enlarge pre-training data scale for better motion understanding and generation capabilities. For fairness, we also evaluate our method without using single-person data pre-training. The performance differences are minimal (FID changes from 1.942 to 2.007, as shown at w/o SP Data in Table 1 and Section 4.3).
> To better understand the impact, we analyze the overlap between datasets in Section 4.4 and present the findings in Figure 3. The analysis shows that the overlap between single- and multi-person data is minimal, meaning the inclusion of single-person data provides only a slight performance boost.
>
> - **Q2. Human evaluation**
>
> Thank you for this valuable suggestion! We conducted a user study using 100 samples from the Inter-X test set to assess user preferences between our TTR model and the SOTA ReGenNet. The results are summarized in Appendix Figure 7 of our revision (Line 722). Our findings show that TTR significantly outperforms ReGenNet in terms of user preferences, particularly when the **action duration is longer**. This further demonstrates the effectiveness of our proposed thinking process, which helps **alleviate accumulated errors**. We consider including this section in our final revision.

---

> ### Author Response · Authors · 2024-11-25
> **Response to Reviewer enFt (2/2)**
>
> Additionally, we have updated our anonymous webpage: Think-Then-React.github.io, where we have included some video cases on user study. If you have further concerns, please don't hesitate to let us know. Your suggestions are valuable to us!
>
> We’d also like to share some positive updates: after thorough discussions, the other three reviewers have provided positive feedback on our responses and revisions, with two of them deciding to increase their ratings. We hope this information may be useful as you continue to evaluate our work.
>
> Thank you again for your time and guidance! We greatly appreciate your help in improving our submission.

---

> > ### Comment · Reviewer_enFt · 2024-11-26
> >
> > I suggest that you include the comparison results with ReGenNet on the the paper. Also, generally speaking, the user study should be part of the main body of the paper, not the appendix. The details of how the user study was set up—such as how you selected the evaluation samples, the number of evaluators, and how you recruited them—should be presented in detail in the paper. I did not see important details about the user study experimental setup.
> >
> > Of course, these are trivial non-technical issues. Thank you for your efforts in addressing my two concerns; I have decided to adjust the rating to 6.

---

> > > ### Author Response · Authors · 2024-11-26
> > >
> > > Thank you for your valuable suggestions and recognition of our work! We will incorporate the user study into the main body of our final manuscript, providing detailed descriptions of experiment setup and discussions as you recommended.

---

### Official Review · Reviewer_De2Y · 2024-11-04

**Soundness:** 2
**Presentation:** 2
**Contribution:** 2
**Rating:** 6
**Confidence:** 4

**Summary:**

The author trains an online complicated framework to generating reaction motion based on the given action motions.  Firstly, the author introduces a unified space-pose representation that effectively converts both absolute space and egocentric pose features into discrete tokens, facilitating LLM training. In addition, the authors introduce the think-then-react framework to get better results.

**Strengths:**

1.  The figure and render visualization is good.
2. The authors introduce a unified space-pose representation that effectively converts both absolute space and egocentric pose features into discrete tokens, facilitating LLM training.

**Weaknesses:**

1. Poor writing. This paper is hard to follow. It's hard for me to distinguish the differences between special words, like “motion”, "pose", "action", "action motion", etc.
2. The training process is complicated. Besides the pertaining and fine-tuning, to improve the training stability, the author also needs to check the validation loss manually and switch the training source.

**Questions:**

1. Could the author provide more video visualization to show the results? It's tough for me to see the generation results from images.
2. Could the author explain further why the thinking part works? i.e. why we need the text as a bridge to connect the input action and output action? From my general understanding, the translation from input action to text has already introduced the information loss.
3. Did 1 FPS and 0.5-second time interval really work? This may lead to heavy leggy and jitter. Can 0.5 seconds really work for an online system?
4. Since the performance depends on the text generation process, could the author try some motion caption benchmarks?

---

> ### Author Response · Authors · 2024-11-22
> **Response to Reviewer De2Y (1/2)**
>
> Thank you for your insightful comments on our work. We have addressed your questions in detail below.
>
> - **Q1. Distinctions among key terminologies**
>
> To improve clarity, we more formally define the terminologies as follows: 1) **Pose** refers to a human body posture or movement within a brief time frame (i.e., timestep), such as "taking one step forward." A pose can be represented as a single token. 2) **Motion** is a sequence of poses, starting with an initial spatial state represented by space features. For example, a motion might be "a person walks three steps." 3) **Action** denotes the input motion initiated by a person, while **reaction** refers to the output motion in response by another person. We have added a paragraph in Section 3.1 to explicitly define these terms (Line 206-209).
>
> We have also refined notations and phrasing throughout the paper particular on Section 3 to enhance clarity and readability.
>
> - **Q2. Training process**
>
> Nice suggestion! The training process is indeed complex as you pointed out. As illustrated in Figure 8 (Appendix Line 722), different tasks converge at different speeds and loss scales: for example, text and space generation tasks tend to converge in much faster speed and smaller loss than motion generation tasks. Thus, to balance different training tasks, we take validation losses after each training epoch as training task sampling weights, to dynamically switch the training source and tasks. For instance, the loss scale of Pose-to-Space (P2S) task is much smaller than Space-to-Pose (S2P), and the P2S task has a much faster converging speed, thus we apply a higher sampling weight on S2P task and a lower sampling rate on P2S. Relevant discussion is highlighted in Section 3.3.1 in our revision (Line 229).
>
> If we have misunderstood your question or the specific meaning of "switching the training sources," please let us know so we can clarify further.
>
> - **Q3. Visualization of results**
>
> Good remark! We have provided qualitative comparisons and failure cases on an anonymous webpage: Think-Then-React.github.io. We have included some of them in our revised paper (Line 432-449).
>
> - **Q4. The necessity of the thinking process**
>
> The necessity of the thinking processes is validated through both experiments and studies for human cognitive process. As shown in Table 1, both the baseline method ReGenNet and our ablation method TTR w/o Thinking perform poorly due to instability and accumulated errors when directly generating reactions from actions.
>
> In the real world, humans typically observe others’ actions, infer their intentions, and then react accordingly. Inspired by this, we designed our method to incorporate a similar reasoning process. Specifically, leveraging the well-annotated Inter-X dataset, which provides fine-grained descriptions for both actors and reactors, we train our model to progress from action captioning to reasoning. This enables the model to utilize the inference capabilities of LLMs in a Chain-of-Thought-like manner for reaction generation. What's more, the reacting process takes **both input action and the predicted prompts from the thinking process**, which introduces more semantics while preserving spatial-temporal details of the input action, mitigating potential information loss.

---

> > ### Author Response · Authors · 2024-11-22
> > **Response to Reviewer De2Y (2/2)**
> >
> > - **Q5. Concerns about FPS settings and online system usability**
> >
> > The 1-FPS setting applies **only to the evaluation model**, which uses low-resolution action inputs for improved distinguishability while keeping reaction outputs at 30 FPS. When more than 1 FPS is applied in evaluation, the metrics have less discriminative power because they pay too much attention to the given input part rather than the generated output motion. When removing the input motion, it is impossible to well understand the semantic of motions. Actually although we sample 1 frame per second, we concatenate the input frame with each frame sampled from output motion to compose complete motions for evaluation. Thus the final metrics can focus on the changes of output motion while keeping the complete semantic of the interactive motions.
> >
> > Regarding the 0.5-second interval, we acknowledge that this might have been unclear in the original description. In our framework, the **Thinking and Reacting processes are parallel** but operate on **different time intervals**. Specifically, TTR performs the thinking process every 0.5 seconds while reacting at every timestep whenever there is new action token input. Between two thinking intervals, TTR uses the results from the most recent thinking process to guide the intermediate reacting steps. We have clarified this in Figure 1. We also have assessed the average inference time per step (AITS) in Section 4.6 (Line 522-526), and confirmed that our method is capable of running in a real-time manner.
> >
> > - Q6. Evaluation on motion caption benchmarks
> >
> > Thank you for this excellent suggestion! We have evaluated TTR on the Inter-X **interaction captioning** benchmark, and the results are presented in the table below. These results demonstrate that our TTR* method effectively translates human interactions into precise captions, outperforming baseline methods.
> >
> > Furthermore, in real-world scenarios, only **partial actions** are often observable, and the ground-truth **reactions are inaccessible**. To test TTR's robustness under such conditions, we evaluate its **action**-to-text generation performance using the first 25%, 50%, and the entire input action. The results indicate that even with only 25% of the input action, TTR accurately predicts the corresponding interaction description, highlighting its strong generalization capability.
> >
> > This is a great suggestion! We plan to include this section in our final revision.
> >
> > |        | RAEs | SeqGan | Seq2Seq | TM2T | TTR* |  TTR | TTR (50%) | TTR (25%) |
> > |--------|:----:|:------:|:-------:|:----:|:----:|:----:|:---------:|:---------:|
> > | Bleu-1 | 28.6 |  45.4  |   53.8  | 56.8 | **60.2** | 55.6 |    54.1   |    52.2   |
> > | Bleu-4 |  9.7 |  14.1  |   18.5  | 21.6 | **25.4** | 20.3 |    18.9   |    16.6   |
> > | Rouge  | 34.1 |  36.8  |   45.2  | 48.2 | **50.5** | 46.4 |    45.3   |    43.0   |

---

> > > ### Comment · Reviewer_De2Y · 2024-11-25
> > > **Discussion with authors**
> > >
> > > Thanks for the clarifications to address my concerns. After reviewing the rebuttal, I decide to improve my score.

---

> > > > ### Author Response · Authors · 2024-11-25
> > > >
> > > > We sincerely thank you for raising your rating score! Your constructive feedback and recognition of our revisions are highly encouraging.

---

### Official Review · Reviewer_X2o2 · 2024-11-04

**Soundness:** 3
**Presentation:** 2
**Contribution:** 3
**Rating:** 6
**Confidence:** 4

**Summary:**

The submission handled the problem of human reaction generation. Specifically, the submission proposed to utilize a unified motion token representation encompassing both egocentric and global spaces. Then, an LLM-based online reaction generation framework TTR is proposed to leverage textual information for training. TTR involves action intention inference as the first step and precise reaction token prediction as the second step. Impressive quantitative improvements are shown.

**Strengths:**

- Explicit decomposition of egocentric poses and global transformations is reasonable, especially considering human reactions.

- Introducing explicit intention modeling as an intermediate stage for reaction generation has been proven helpful.

**Weaknesses:**

- The overall writing clarity could be improved. Please refer to Questions.

- The evaluation settings could be ambiguous. Please refer to Questions.

- The effectiveness of the proposed unified motion tokenizer is not specifically discussed in experiments. Is it feasible to treat TTR w/o ALL P.T. as MotionGPT w/ unified representation? To understand this, details on how MotionGPT is adapted to multi-person scenarios should be provided.

- Despite the considerable quantitative improvements, the improved quantitative metrics are essentially indirect proxies of the real reaction generalization performance. A qualitative comparison of how the proposed method outperforms previous SOTAs is still expected. Animated motions would be a better form of visualization.

- Moreover, a failure case study is also expected if possible.

**Questions:**

- In L219, it is unclear how a pair of persons should face the positive z-axis. It is more understandable that the reactee faces the positive z-axis.

- In L221, are (x,z) corresponding to the pelvis 2D coordinates?

- Terms in L231 are inconsistent, making it hard to understand.

- Why the evaluation model from Inter-X is not adopted, which has a high ACC. over 90% compared to the 46.3% of the adopted model?

---

> ### Author Response · Authors · 2024-11-22
> **Response to Reviewer X2o2 (1/2)**
>
> We sincerely appreciate your valuable comments and recognition of our work. Below, we provide detailed responses to your questions.
>
> - **Q1. Effectiveness of unified tokenizer and settings of MotionGPT**
>
> Thank you for your insightful suggestion. To directly assess the effectiveness of our unified tokenizer, we have evaluated a **plain** tokenizer (i.e., a **non-decoupled motion tokenizer** that directly encodes multi-person motion) on motion reconstruction and action-to-reaction generation (i.e., use the tokens encoded by the plain tokenizer for LLM training and inference). The results in the table below indicate that the plain tokenizer fails to accurately reconstruct multi-person motion, leading to ambiguous motion-text understanding during training and inference. This hinders LLM's ability to effectively perform the thinking-then-reacting process, resulting in deteriorated reaction generation performance. We consider including this analysis in the final revision.
>
> Regarding MotionGPT, we extend the motion tokenizer of MotionGPT to encode multi-person motion while keeping other settings unchanged. However, TTR w/o ALL PT is not equivalent to MotionGPT w/ Unified Representation. This is because we propose a thinking-then-reacting mechanism that fully leverages the inference capabilities of LLMs to mitigate accumulated errors in action-to-reaction generation. We added adaptation details of MotionGPT for multi-person scenarios to Section 4.1 (Line 365).
>
> |Method||FID|Top-1|Acc.|
> |--|--|--|--|--|
> |Plain|Recon.|0.983±.000|0.392±.000|0.254±.000|
> ||Gen.|5.919±.012|0.205±.003|0.141±.003|
> |Ours|Recon.|0.262±.000|0.501±.000|0.395±.000|
> ||Gen.|**1.942±.017**|**0.423±.005**|**0.318±.003**|
> ||Gen. w/o Thinking|3.828±.016|0.367±.003|0.230±.036|
> ||Gen. w/o ALL PT|3.467±.113|0.398±.007|0.288±.002|
>
> - **Q2. A pair of persons or the reacteee facing the positive z-axis?**
>
> You are right. Only the reactee is aligned to face the positive z-axis when we reposition the reactee at the origin of the absolute coordinate system. This adjustment is done while preserving the relative spatial relationship between the two persons. Then the absolute spatial features of each individual, including their x and z positions and body orientation r, are extracted as space tokens.  As shown in Figure 2(a), the reactee’s absolute spatial feature is represented as (x0,z0,r0), meaning they are positioned at the origin (x0,z0) and oriented to face the positive z-axis (i.e., r0). The other person's spatial feature is (x2,z5,r6), indicating their position is away from the origin and their orientation differs from the reactee's.
>
> - **Q3. Meaning of (x, z)**
>
> Yes, the (x, z) values indeed correspond to the 2D coordinates of the **pelvis**. These values represent the position of the pelvis in the horizontal plane, which serves as the reference point for defining spatial positions and relationships. We have explicitly stated this in the revised version (Line 216).
>
> - **Q4. Inconsistent terms**
>
> Thank you for pointing out the inconsistency. We have carefully reviewed and revised the terminology in Section 3.1 to ensure consistency throughout the text (Line 227). We have also clarified and aligned other inconsistencies in our revision and highlighted them.
>
> - **Q5. Why is the Inter-X evaluation model not adopted**
>
> The main reason is the **difference between motion representations**. To ensure computational efficiency, previous works [1,2] mainly adopt a standardized motion representation comprising skeleton positions, velocities, and orientations, amounting to approximately 262 dimensions. We follow this widely used motion representation. In contrast, Inter-X’s evaluation model utilizes a **denser motion representation, SMPL-X**, a 3D human body model that includes facial expressions and is based on skinning and blend shapes, with more than thousands of dimensions. Due to these differences in motion representations, we cannot directly adopt the Inter-X classification model.
>
> Previous works [1,2] based on the same representations lack interaction labels, which is necessary for calculating accuracy. That's why we re-train an evaluation model.
>
> Regarding the lower accuracy, there are two main reasons: 1) Inter-X employs a GNN-based evaluation model specifically optimized for classification tasks. In contrast, our evaluation model should **balance between classification accuracy and ranking metrics** (e.g., R-Precision). To achieve this, we adopt a CLIP-like training strategy, similar to [2]. 2) Our focus is on evaluating the quality of generated reactions. To this end, we mask a large proportion of the input action sequence to enhance the model’s **discriminative power**. While this improves reaction quality assessment, it comes at the expense of input detail and classification accuracy.

---

> ### Author Response · Authors · 2024-11-22
> **Response to Reviewer X2o2 (2/2)**
>
> - **Q6. Qualitative comparisons and failure cases**
>
> Good remark! We have provided qualitative comparisons and failure cases on an anonymous webpage: Think-Then-React.github.io. We have included some of them in our revised paper (Line 432-449).
>
> [1] Chuan Guo, Shihao Zou, Xinxin Zuo, Sen Wang, Wei Ji, Xingyu Li,  and Li Cheng. Generating diverse and natural 3d human motions from text.  CVPR 2022
>
> [2] Han Liang, Wenqian Zhang, Wenxuan Li, Jingyi Yu, and Lan Xu.  Intergen: Diffusion-based multi-human motion generation under complex  interactions. IJCV 2024

---

> > ### Comment · Reviewer_X2o2 · 2024-11-23
> >
> > Thanks! I appreciate the responses, which help address my concerns. There are some further questions.
> >
> > **On Q1.** According to the shared results, it seems that TTR is highly reliable on the decoupled tokenizer, as TTR+plain seems to be worse than most baselines, which could be counter-intuitive to some extent. Some details on the adopted plain tokenizer would be preferred. Is it sharing the same architecture and hyperparameters as the decoupled tokenizer? How the multi-person motion is directly represented, like including the (x, z, r) in the VQ-VAE?
> >
> > **On Q5.** Though masking input action sequences is an intuitive design choice to encourage the model to focus on the generated reaction, it might tune down the focus on the action-reaction correspondence. To my current knowledge, I'm not sure which is more important for human reaction generation evaluation between semantic-reaction correspondence and action-reaction correspondence, while I prefer the latter slightly more.
> >
> > **On the added user study and visualization.** I appreciate the newly included user study and visualizations. Is it possible to provide videos from different viewpoints of the sample ``Two people stand facing each other. The first person leans towards the second person, and then the second person taps her/him``? Also, it would be appreciated if some representative user-study samples, including both the winning and losing samples of TTR, could be shared with user reviews.

---

> > > ### Author Response · Authors · 2024-11-23
> > >
> > > Thank you for your insightful questions! We will prepare relevant materials and provide a response within 24 hours.

---

> > > ### Author Response · Authors · 2024-11-24
> > > **Response to Reviewer X2o2 (2/2)**
> > >
> > > - **On user study and visualization**:
> > >
> > > Thank you for your careful feedback on our visualizations and user study. Upon switching the camera perspective in the renderer, we observed that, in the "bow-tap" case you mentioned, the reactor does not accurately tap the actor. This case should therefore be categorized as a failure case, and we sincerely appreciate your observation.
> > >
> > > In response to your comment, we have updated all video examples with a **moving camera** to reduce occlusion and provide clearer views from different angles. Additionally, we have included some representative user study samples, including both successful and failure cases. These updates are now available on our revised webpage. If the page does not appear updated, please wait a few minutes and refresh your browser cache.
> > >
> > > ---
> > > Once again, we truly appreciate your insightful suggestions, which not only helped improve our current work but also pointed out promising directions for future research.

---

> > > > ### Comment · Reviewer_X2o2 · 2024-11-25
> > > >
> > > > Thanks for the detailed responses! I'm especially impressed with the comprehensive analysis of the evaluation model, which is inspiring with helpful insights. I encourage the authors to include the analyses in the appendix of the final version. And I would increase my ratings.

---

> ### Author Response · Authors · 2024-11-24
> **Response to Reviewer X2o2 (1/2)**
>
> - **On Q1: About our unified tokenizer**
>
> Without a unified motion representation enabled by our decoupled tokenizer, the performance of TTR deteriorates significantly due to **much lower-quality training data**, which is critical for LLMs. When using the plain tokenizer, we observe a more unstable training process. Worse still, TTR with the plain tokenizer often fails to predict a meaningful sentence during the Thinking stage, **disabling our Think-Then-React framework**. This results in the poorest performance among our ablation studies (comparable only to MotionGPT).
>
> Regarding the details of the plain VQ-VAE, your understanding is correct. The plain tokenizer shares the same architecture and hyperparameters as the decoupled tokenizer.
>
> We would like to clarify the details of the VQ-VAE with an example on our data processing pipeline, and we have also added **Figure 9** at Line 774 of Appendix to illustrate this process in detail: First, we **normalize the reaction** location and orientation (to the original point and facing z-positive axis), e.g., from (x=1, z=2, r=45°) to (x=0, z=0, r=0). Simultaneously, the action’s location and orientation are adjusted to maintain their relative position and orientation, e.g., from (x=3, z=4, r=225°) to (x=2, z=2, r=180°). Then, for **reaction**, we directly encode it with VQ-VAE into pose tokens, which is already normalized, e.g., <p2><p5><p6>.
>
> For **action**:
>
> - With our decoupled tokenizer: We **normalize** it to (x=0, z=0, r=0) and then encode it with VQ-VAE to obtain pose tokens, e.g., <p1><p2><p8>. The final action sequence could be represented as <x2><z2><r180><p1><p2><p8>.
>
> - With the plain tokenizer: We **retain the unnormalized space features (x=2, z=2, r=180°)**, maintaining action-reaction relative position and orientation, and then encode it with VQ-VAE to obtain pose tokens, e.g., <p10><p21><p33>. The final action sequence could be represented as <p10><p21><p33> without space tokens, as the spatial features are implicitly embedded within these tokens.
>
> Although the unnormalized action features theoretically combine spatial and motion information, encoding such features directly with VQ-VAE often leads to inefficient use of the codebook with **limited size**. For instance, the **same motion** sequence at **different positions** or orientations would be encoded as **different token sequences**, whereas our decoupled tokenizer only modifies the **prefix** of the sequence.
>
> We also evaluated the **perplexity (PPL)** of the two codebooks (both codebook size is 256), and found that our decoupled tokenizer converges at **139.9**, enabling higher precision in diverse motion representation, while the plain tokenizer achieves only **94.2**, indicating underutilization or even collapse of the codebook.
>
> - **On Q5: Balancing semantic-reaction and action-reaction correspondence**
>
> Thank you for your insightful question. In brief, our findings suggest that **the 1 FPS setting benefits both types of correspondence, although there may be more efficient solutions to explore**.
>
> The results in Figure 5 demonstrate the effectiveness of the 1 FPS setting in improving semantic-reaction correspondence. To further investigate its impact on action-reaction correspondence, we conducted an additional experiment: we introduced random **spatial or temporal shifts** to the ground-truth reaction, **disrupting the action-reaction correspondence**, and evaluated the sensitivity of our metrics (FID and ranking scores) to these shifts. We compared our 1 FPS setting against the default setting, which uses the entire action-reaction sequence as input. The results, expressed as "ours/default," are presented in the table below:
>
> |  |  | Time Shift (Seconds) |  |  | Space Shift (Meters) | |
> |--|:--:|:--:|:--:|:--:|:--:|:--:|
> | | 0.25      | 0.5      | 1       | 0.05           | 0.1       | 0.2            |
> | FID Difference     | **0.832**/0.725 | **0.961**/0.891 | 1.279/1.281     | **0.745**/0.646 | **0.761**/0.657 | **0.843**/0.703 |
> | Ranking Difference | **0.049**/0.009 | **0.076**/0.056 | **0.194**/0.165 | **0.009**/0.003 | **0.027**/0.010 | **0.064**/0.037 |
>
> These findings indicate that, even without explicitly optimizing for action-reaction correspondence during training, our 1 FPS setting demonstrates **greater discriminative power** and is **more sensitive to shifts** that degrade action-reaction correspondence. We mainly attribute this to: 1) Action dominates overall semantics of interaction. As a result, feeding the entire action as input might lead to **shortcut learning** by the evaluation model, ignoring both semantic-reaction and action-reaction correspondence. 2) Even with a high masking ratio applied to the input action, the remaining sequences retain **enough semantic and spatial-temporal details** to guide accurate evaluation. Like videos, motion sequences are relatively low in information density compared to other modalities like texts.

---

> ### Author Response · Authors · 2024-11-25
>
> We sincerely appreciate your recognition to our work! We will include detailed analysis in the appendix of the final version. Your support and increased rating mean a great deal to us!

---

### Author Response · Authors · 2024-11-22
**Global Response**

Dear AC and reviewers,

We sincerely thank the reviewers for their valuable and constructive feedback on our submission. We are encouraged by the recognition of our work’s contributions: **All reviewers** acknowledged the innovation of our decoupled space-pose tokenizer for effective multi-person motion representation. Reviewers **X2o2** and **wKay** highlighted the novelty of our explicit thinking-then-reacting framework. Reviewers **enFt** and **wKay** appreciated the strong performance of our overall method. Reviewer **De2Y** commended the quality of our illustrations. Reviewer **enFt** emphasized the importance of the issue we address. Reviewer **wKay** recognized the soundness and effectiveness of our training strategy.


We have responded individually to each reviewer to address their questions. Below, we provide a concise summary of our responses to some common concerns:

- Video Visualizations: We have provided qualitative comparisons and examples of failure cases in videos hosted on an anonymous webpage: Think-Then-React.github.io. Additionally, we have included some of the visualizations in the revised paper.
- Writing Issues: We appreciate the reviewers’ feedback regarding writing issues, including improper notation usage, unclear definitions of key concepts, and other errors. We have carefully considered these suggestions and revised the paper accordingly.
- Key Terminologies: To improve clarity, we more formally define the terminologies as follows: 1) Pose refers to a human body posture or movement within a brief time frame (i.e., timestep), such as "taking one step forward." A pose can be represented as a single token. 2) Motion is a sequence of poses, starting with an initial spatial state represented by space features. For example, a motion might be "a person walks three steps".


In response to your comments and suggestions, we have carefully revised the manuscript and marked all changes in blue for ease of review. Below is a summary of the key updates:

In the Main Paper:

- Figure 1: Updated to better illustrate our framework, along with a revised caption.
- Section 3.1: Notations have been aligned, and we clarified the terminologies pose and motion.
- Section 3.2: Added details on space and pose representations and addressed inconsistent notations.
- Section 3.3: Clarified the pre-training task details and improved the description of the training data sampling strategy.
- Figure 4: Added visualized examples, including a failure case.

In the Appendix:

- Section A.3: Included experiments on motion captioning.
- Section A.4: Added ablation studies on the thinking process.


Best regards,

Authors of submission 13835

---

### Meta-Review · Area_Chair_ayUd · 2024-12-20

**Metareview:**

Summary: The paper investigates a novel method Think-Then-React (TTR) for action-to-reaction motion generation in two-person interactions. TTR leverages a unified motion token representation that encodes both absolute (global) space and egocentric (local) pose features into discrete tokens, enabling effective training with LLMs. It employs a two-step process: action intention inference and precise reaction token prediction. By utilizing a pre-trained LLM to understand and generate both motion and textual descriptions, the method enhances coherence in the generated reactions. Quantitative experiments demonstrate that TTR outperforms state-of-the-art techniques, achieving notable improvements on the Inter-X dataset.

Strengths and Weaknesses: The reviewers generally recognize that this paper addresses a crucial problem in human interactions –  improving action-to-reaction motion generation. They also appreciate the decomposition of reaction generation into a thinking process using LLMs and a reacting process with VQ-VAE, the unified space-pose token representation, and strong performance of the proposed approach.

However, they express reservations about 1) the absence of qualitative results, such as demo videos and user study-based evaluations; 2) potential unfair comparisons due to the use of mixed datasets without adequate clarification; 3) unclear and inconsistent terminology throughout the paper; 4) ambiguous evaluation settings, including insufficient explanation of the metrics used; 5) the complexity of the training process; and 6) the lack of a convincing rationale for certain design choices.

Discussion and Recommendation: The authors successfully addressed most of these points during the discussion phase, by providing detailed clarifications and additional experimental results, and all reviewers recommend acceptance.

The area chair concurs with the reviewers and recommends that the authors incorporate the clarifications, explanations, analyses, and results presented or promised during the discussion phase (such as the detailed setup of the user study) into the camera-ready version.

**Additional Comments On Reviewer Discussion:**

The authors successfully addressed most of these points during the discussion phase, by providing detailed clarifications and additional experimental results, and all reviewers recommend acceptance.

The area chair concurs with the reviewers and recommends that the authors incorporate the clarifications, explanations, analyses, and results presented or promised during the discussion phase (such as the detailed setup of the user study) into the camera-ready version.

---

### Decision · Program_Chairs · 2025-01-22

Accept (Poster)